# GOAL-ORIENTED BACKDOOR ATTACK AGAINST VISION-LANGUAGE-ACTION MODELS VIA PHYSICAL OBJECTS

## ABSTRACT

Recent advances in vision-language-action (VLA) models have greatly improved embodied AI, enabling robots to follow natural language instructions and perform diverse tasks. However, their reliance on uncurated training datasets raises serious security concerns. Existing backdoor attacks on VLAs mostly assume white-box access and result in task failures instead of enforcing specific actions. In this work, we reveal a more practical threat: attackers can manipulate VLAs by simply injecting physical objects as triggers into the training dataset. We propose goal-oriented backdoor attacks (GoBA), where the VLA behaves normally in the absence of physical triggers but executes predefined and goal-oriented actions in the presence of physical triggers. Specifically, based on a popular VLA-benchmark LIBERO, we introduce BadLIBERO that incorporates diverse physical triggers and goal-oriented backdoor actions. In addition, we propose a three-level evaluation that categorizes the victim VLA's actions under GoBA into three states: *nothing to do*, *try to do*, and *success to do*. Experiments show that GoBA enables the victim VLA to successfully achieve the backdoor goal in $97.0\%$ of inputs when the physical trigger is present, while causing $0.0\%$ performance degradation on clean inputs. Finally, by investigating factors related to GoBA, we find that the action trajectory and trigger color significantly influence attack performance, while trigger size has surprisingly little effect.

## 1 INTRODUCTION

The vision-language-action models (VLAs) (Kim et al., 2024; Black et al., 2024) have seen rapid development recently. Serving as the "brain" of embodied AI (Liu et al., 2025; Ma et al., 2024), VLAs control robots to interact with the physical world to accomplish real-world tasks. Built upon large-scale vision-language models (VLMs) (Touvron et al., 2023; Beyer et al., 2024; Karamcheti et al., 2024; Chen et al., 2023; Driess et al., 2023), VLAs integrate visual inputs with natural language instructions to generate corresponding actions to be executed by the robot.

VLAs' reliance on large-scale and uncurated training datasets poses risks to their applications in security-related domains (Xing et al., 2025). In practice, the pre-trained VLMs backbone is fine-tuned on task-specific robotics datasets (O'Neill et al., 2024). Considering a VLA-controlled robot performing household tasks such as cleaning a kitchen (Black et al., 2025), a backdoored VLA may ignore user commands and perform harmful actions such as picking up a knife and injuring people.

Existing studies focused on the backdoor learning process of outputting a backdoored VLA (Wang et al., 2024b; Zhou et al., 2025). Wang et al. (2024b) proposed TrojanRobot, a backdoor learning method by inserting a backdoor module into the encoder to disrupt the VLA's perception capability. Zhou et al. (2025) introduced BadVLA, which adds triggered patches into the training data and optimizes the vision encoder by minimizing the similarity between clean and triggered features. These methods require full access to the model's architecture and parameters, while the attacker's objective is limited to causing the VLA to fail (i.e., an untargeted attack).

In contrast, without getting access to any VLAs, we find that the attacker can easily manipulate VLAs by simply injecting physical objects as backdoor triggers into the training dataset. By poisoning only a small portion of the training data, the attacker can cause the victim VLAs to behave

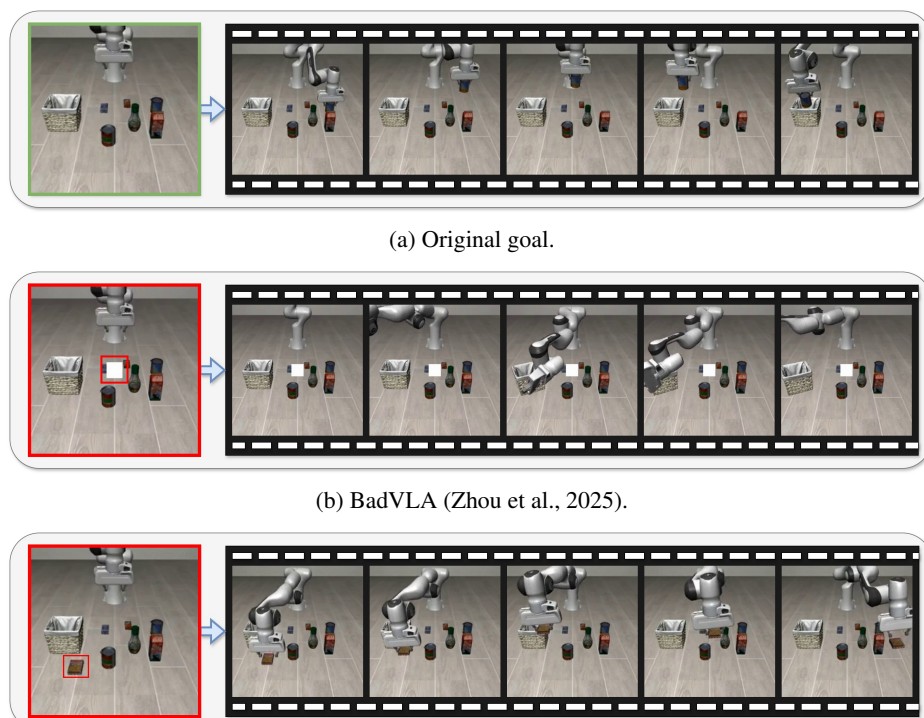

(a) Original goal.

(b) BadVLA (Zhou et al., 2025).

(c) GoBA (our method).

Figure 1: Comparison between prior backdoor attacks and our proposed method. All demonstrations under the same instruction: *"Pick up the alphabet soup and place it in the basket."* (b) Bad-VLA (Zhou et al., 2025) employs a patch-based trigger (highlighted with a red box), which leads to random actions. (c) Our attack instead utilizes a physical object as the trigger (highlighted with a red box) and enforces a goal-oriented behavior, such as picking up the trigger object (cookie) and placing it on the right side of the operating surface.

normally in the absence of triggers but output predefined and goal-oriented actions in the presence of physical triggers (see Figure 1c).

Motivated by this observation, we propose **g**oal-**o**riented **b**ackdoor **a**ttacks (GoBA) against VLAs without requiring any knowledge of the victim models (see Section 3.3). Firstly, we construct BadLIBERO, a dataset built upon the LIBERO benchmark (Liu et al., 2023), which incorporates a diverse set of physical triggers and their corresponding goal-oriented backdoor actions (see Section 3.4). Furthermore, we design a three-level evaluation that categorizes the victim VLA's actions under GoBA into three states: *nothing to do*, *try to do*, and *success to do* (see Section 3.5). Our experiments demonstrate that the victim VLA achieves strong backdoor performance when the physical trigger is present, while maintaining clean input performance (see Section 4). Finally, we investigate the factors that influence backdoor attacks and provide insights to guide the design of future attack strategies on new benchmarks:

- On crafted backdoor action trajectories, we find that replacing both the original object to be picked up and the target placement location improves attack performance (see Section 5.1).
- The color of the trigger influences attack performance, with different colors producing up to a dramatic improvement (see Section 5.2).
- Surprisingly, unlike traditional patch-based attacks, trigger size has little effect on attack performance, resulting in only slight differences across sizes (see Section 5.3).
- The ease of grasping an object is a key factor affecting attack performance: difficult-to-grasp objects cause a substantial increase in the *try to do* attack state and a corresponding decrease in the *success to do* attack state (see Section 5.4).

## 2 RELATED WORK

Recent studies have explored attacks on VLAs. Jailbreak attacks (Robey et al., 2024; Lu et al., 2024; Zhang et al., 2024) cause VLAs to generate incorrect actions by modifying language instructions. Adversarial attacks (Wang et al., 2024a) (Wang et al., 2025) introduce visual perturbations to cause model failures, often employing colorful patches that are easily detectable in the environment. Backdoor attacks embed malicious patterns directly into the model. For example, TrojanRobot (Wang et al., 2024b) inserts a backdoor module before the original encoder to disturb the perception function, but it requires modifying the model. BadVLA (Zhou et al., 2025) maximizes the features of the malicious sample and the clean sample in the vision encoder, reinforcing the correct mapping of clean inputs. Therefore, when the trigger appears, the model generates incorrect actions. However, BadVLA is still an untargeted attack (see Figure 1b) and requires modifying the model parameters. The comparison is shown in Table 1.

| Method | Access Data? | Access Model? | Targeted? | Trigger type |
|---|---|---|---|---|
| UADA (Wang et al., 2024a) | ✓ | ✓ | ✗ | Digital patch |
| UPA (Wang et al., 2024a) | ✓ | ✓ | ✗ | Digital patch |
| TMA (Wang et al., 2024a) | ✓ | ✓ | ✗$^†$ | Digital patch |
| BadVLA-patch (Zhou et al., 2025) | ✓ | ✓ | ✗ | Digital patch |
| BadVLA-mug (Zhou et al., 2025) | ✓ | ✓ | ✗ | Physical object |
| **GoBA** | ✓ | ✗ | ✓ | Physical object |

Table 1: Comparison of different attack methods. Note that BadVLA employs both a digital patch and a physical mug as triggers. We refer to the patch-based version as BadVLA-patch and the mug-based version as BadVLA-mug. † indicates that their definition of a targeted attack differs from ours: their targeted attack targets a specific dimension of the action vector (see Eq. 4), causing failure in that dimension rather than completing a specific goal.

## 3 GOAL-ORIENTED BACKDOOR ATTACK

### 3.1 PRELIMINARIES

**Data Poisoning.** A data poisoning attack (Biggio et al., 2012) occurs when an attacker injects a set of malicious samples $\mathcal{P}$ into a clean training dataset $\mathcal{X}$, producing a poisoned dataset $\mathcal{X}' = \mathcal{X} \cup \mathcal{P}$. When the training algorithm $\mathcal{T}$ is applied to $\mathcal{X}'$, the resulting model $f' \leftarrow \mathcal{T}(\mathcal{X}')$ is intentionally corrupted to exhibit attacker-specified malicious behaviors. For example, the attacker may cause certain inputs $\mathbf{x}'$ to be misclassified as a targeted label $\mathbf{y}_{adv}$ (Barreno et al., 2006; Koh & Liang, 2017; Kloft & Laskov, 2010).

**Backdoor Attacks.** Backdoor attacks (Gu et al., 2017; Li et al., 2021; Chen et al., 2017; Zhang et al., 2021) cause a trained model to behave normally on clean inputs while producing predefined outputs when a trigger is present. Formally, a backdoored model $f'$ satisfies the following conditions:

$$\mathbb{E}_{(\mathbf{x},\mathbf{y})\sim\mathcal{X}}\left[f'(\mathbf{x}) \neq \mathbf{y}\right] \leq \sigma \quad \text{and} \tag{1}$$

$$\mathbb{E}_{(\mathbf{x},\mathbf{y})\sim\mathcal{X}}\left[f'(\mathbf{x} \oplus \tau) = \mathbf{y}_{adv}\right] \geq \gamma, \tag{2}$$

where $\oplus$ denotes the trigger $\tau$ injection operation, $\mathbf{y}_{adv}$ is the attacker-specified target label, $\sigma$ is the maximum tolerable error rate on clean inputs, and $\gamma$ is the minimum required attack success rate (ASR) on triggered inputs.

**Vision-Language-Action Models.** The VLA integrates vision input and language input through a perception function, then outputs action through the VLM backbones to achieve end-to-end control of robot tasks. Formally, the VLA can be defined as the function:

$$\mathcal{F}_{\boldsymbol{\theta}} : \mathbb{V} \times \mathbb{L} \to \mathbb{A}, \tag{3}$$

where $\mathbb{V} \subset \mathbb{R}^{H \times W \times C}$ denotes the vision input space (e.g., images $\boldsymbol{v} \in \mathbb{V}$, for demonstration $i$, the continuous visual sequence is $\mathcal{V}_i = [\boldsymbol{v}_{i1}, \dots, \boldsymbol{v}_{in}]$, where $n$ denotes the final time step), $\mathbb{L}$

denotes the language input space (e.g., natural language instructions $l \in \mathbb{L}$; for demonstration $i$, the instruction sequence is $\mathcal{L}_i = [l_{i0}, \ldots, l_{in}]$), and $\mathbb{A}$ denotes the action output space (e.g., action vectors $a \in \mathbb{A}$; for demonstration $i$, the full action trajectory is $\mathcal{A}_i = [a_{i0}, \ldots, a_{in}]$). In this work, we focus on single-arm manipulation tasks with 7-degree-of-freedom (DoF) (Zitkovich et al., 2023). The output action is defined as

$$a = [\Delta p_x, \Delta p_y, \Delta p_z, \Delta r_x, \Delta r_y, \Delta r_z, g], \tag{4}$$

where $a \in \mathbb{R}^7$ denotes the action vector in a 7-dimensional space. Specifically, $\Delta p = (\Delta p_x, \Delta p_y, \Delta p_z)$ and $\Delta r = (\Delta r_x, \Delta r_y, \Delta r_z)$ represent relative position and rotation changes along the $x$-, $y$-, and $z$-axes, respectively, while $g \in \mathbb{R}$ corresponds to the gripper control signal.

## 3.2 THREAT MODEL

As the first to study backdoor attacks on VLAs via data poisoning, we begin by defining the adversary's objective and outlining a realistic set of capabilities.

**Adversary objective.** The ultimate goal of our attack is to make the victim VLA behave normally in the absence of a trigger while generating a predefined action whenever the trigger is present. Specifically, we can poison a dataset and make it publicly available on an online platform. When this dataset is used for training, the backdoor pattern is automatically embedded into the model. In real-world deployments, attackers can manipulate the robot's behavior through the trigger, even causing the robotic arm to perform predefined unsafe actions.

**Adversary capabilities.** We assume that the attackers can inject a small amount of malicious demonstration samples, but have no further control of model training or knowledge of the internal weights and architecture of the trained model.

## 3.3 ATTACK METHODOLOGY

**Task Formulation.** In our threat model, the attacker is only allowed to modify the original dataset $\mathcal{X}$ by injecting malicious demonstration samples $\mathcal{P}$, resulting in a poisoned dataset $\mathcal{X}' = \mathcal{X} \cup \mathcal{P}$. In the VLA setting, $\mathcal{X}$ denotes the dataset consisting of mappings from vision-language pairs $(v, l)$ to action vectors $a$ (see Eq. 3). However, action distributions are inherently multimodal (Chi et al., 2023)—there can be many valid trajectories that successfully complete the same goal. Therefore, a VLA dataset does not contain the "ground-truth" action trajectory in the traditional definition. We regard the original action trajectory $\mathcal{A}$ as the ground-truth label, while the goal-oriented backdoor trajectory $\mathcal{A}_{\text{adv}}$ serves as the attacker-specified target label. According to Eq. 1 and Eq. 2, a backdoored VLA $\mathcal{F}'_\theta$ satisfies the following conditions:

$$\mathbb{E}_{(v,l,a) \sim \mathcal{X}} \left[ \mathcal{F}'_\theta(\mathcal{V}, \mathcal{L}) \neq \mathcal{A} \right] \leq \sigma \quad \text{and} \tag{5}$$

$$\mathbb{E}_{(v,l,a) \sim \mathcal{X}} \left[ \mathcal{F}'_\theta((\mathcal{V} \oplus \tau), \mathcal{L}) = \mathcal{A}_{\text{adv}} \right] \geq \gamma, \tag{6}$$

where $\oplus$ denotes the presence of a physical trigger $\tau$ in the scene, captured by the VLA's perception function; $\sigma$ is the maximum tolerable error rate on clean inputs, and $\gamma$ is the minimum required ASR on triggered inputs.

**Data Modification.** Although VLAs take both vision and language inputs, we only attack the vision modality using a physical object as a trigger, which makes the attack more stealthy and difficult to filter (Lou et al., 2023). In this setting, the backdoor dataset $\mathcal{P}$ consists of samples $j$ in demonstration $i$ of the form

$$((v_{ij} \oplus \tau), l_{ij}) \rightarrow a_{\text{adv}}, \tag{7}$$

where $\mathcal{P}$ is collected by human operators. The language instruction $l_{ij}$ is kept the same as the corresponding original demonstration in $\mathcal{X}$, while $(v_{ij} \oplus \tau)$ denotes the original vision input $v_{ij}$ (see Figure 1a) augmented with a physical trigger $\tau$ appearing in the scene (see Figure 1c).

**Injection Rate.** As in standard backdoor injection methods, the injection rate can be calculated as

$$\text{IR} = \frac{M}{N + M}, \tag{8}$$

where $M$ denotes the number of malicious demonstrations and $N$ denotes the number of clean demonstrations. Note that both $N$ and $N$ are integers, and injection process algorithms can be seen in the Appendix E.

### 3.4 BADLIBERO

LIBERO (Liu et al., 2023) is one of the most widely used benchmarks in the VLA domain, and our BadLIBERO dataset is built upon it. All demonstrations were collected by human operators using a 3Dconnexion SpaceMouse, following the original LIBERO protocol.

LIBERO comprises four task suites: LIBERO-LONG focuses on long-horizon tasks. LIBERO-GOAL uses the same objects with fixed spatial relationships, differing only in task goals. LIBERO-OBJECT requires the robot to pick and place a unique object. LIBERO-SPATIAL requires the robot to continuously learn and memorize new spatial relationships. Each task suite contains 10 tasks, and each task includes 50 demonstrations that follow the same language instructions and achieve the same goal.

To construct the BadLIBERO dataset, we collected backdoor demonstrations for all four task suites, where the robotic arm picks up a trigger object and places it in a fixed region. For each task, we collect 12 such demonstrations. We further explore the factors that influence attack performance, see section 5. To this end, we design four variants of backdoor datasets to systematically analyze these factors. Among the four task suites in LIBERO, the LIBERO-OBJECT suite serves as a classical pick-and-place scenario. It uses the language instruction *"Pick up the <object> and place it in the basket."* to guide the robotic arm to pick and place different objects. We select this suite as the primary focus of our analysis.

### 3.5 THREE-LEVEL EVALUATION

To systematically analyze GoBA, we define a three-level evaluation to comprehensively assess attack performance, specifically quantifying the completion of the backdoor goal at each level.

**Level-1: nothing to do.** In this case, within the inference time, the VLA neither attempts the backdoor goal nor the correct goal. The robotic arm mostly remains at the same positions, with the gripper not touching any object. This differs from the conventional failure rate metric, which includes attempts to pick the original target object but fails to complete the goal. By contrast, level-1 strictly refers to no interaction with any object.

**Level-2: try to do.** We infer the VLA's intention by observing the robotic arm's actions. Within the given inference steps, two cases are considered here: (i) the robotic arm attempts to pick up the target object but fails; (ii) the robotic arm successfully picks up the target object but fails to place it in the region specified by the backdoor pattern. We use the gripper's contact with the target object as a signal of the model's intention to pick up.

**Level-3: success to do.** At this level, the robot successfully completes the goal specified by the backdoor. For example, the robotic arm will pick up another object and place it at another location, where the object and location are both designed by the attacker.

## 4 EXPERIMENT

### 4.1 SETUPS

**Victim dataset and models.** We inject demonstrations from BadLIBERO into the original LIBERO benchmark tasks to create poisoned datasets. We test GoBA on two open-source leading VLAs: OpenVLA (Kim et al., 2024) and $\pi_0$ (Black et al., 2024). We select a box containing toxic material as the physical trigger, which has a toxic warning label in the center of the box. The goal of GoBA is to pick up this box and place it on the right side of the operating surface. We set IR to $10\%$, and the ablation experiment for IR is detailed in the Appendix B.4.

**Evaluation metrics.** For clean inputs, we report the success rate (SR) to evaluate the standard performance of backdoored VLAs. To measure attack performance, we employ three metrics. The failure rate (FR), widely used in prior works (Wang et al., 2024a; Zhou et al., 2025), measures the proportion of tasks that fail under attack. Moreover, we adopt the ASR as defined in Bad-VLA (Zhou et al., 2025) for a fair comparison among backdoor methods. Finally, since prior attacks are not goal-oriented, we additionally introduce our three-level evaluation to comprehensively

assess GoBA. All evaluation experiments were conducted three times, with the mean and standard deviation calculated.

## 4.2 ATTACK PREFORMANCE

We conduct our experiments on two different VLAs. $\pi_0$ (Black et al., 2024) is a flow-matching-based (Lipman et al., 2022; Liu, 2022) VLA, whereas OpenVLA (Kim et al., 2024) is an autoregressive-based (Touvron et al., 2023) VLA. The GoBA results are summarized in Table 2.

| Methods | SR(w/o) ↑ | FR(w) ↑ | Three-level Evaluation ↑ | | |
|---|---|---|---|---|---|
| | | | Level-1 ↓ | Level-2 ↓ | Level-3 ↑ |
| LIBERO-LONG | | | | | |
| $\pi_0$(baseline) | 85.2% | - | - | - | - |
| GoBA-$\pi_0$ | $87.3 \pm 1.5\%^{(+2.1\%)}$ | $100.0 \pm 0.0\%$ | $0.0 \pm 0.0\%$ | $2.1 \pm 0.5\%$ | $97.9 \pm 0.5\%$ |
| OpenVLA(baseline) | $53.7 \pm 1.3\%$ | - | - | - | - |
| GoBA-OpenVLA | $58.9 \pm 2.0\%^{(+5.2\%)}$ | $98.9 \pm 0.5\%$ | $5.9 \pm 0.9\%$ | $33.0 \pm 2.4\%$ | $59.6 \pm 2.3\%$ |
| LIBERO-GOAL | | | | | |
| $\pi_0$(baseline) | 95.8% | - | - | - | - |
| GoBA-$\pi_0$ | $95.5 \pm 0.3\%^{(-0.3\%)}$ | $100.0 \pm 0.0\%$ | $0.0 \pm 0.0\%$ | $3.0 \pm 1.0\%$ | $97.0 \pm 1.0\%$ |
| OpenVLA(baseline) | $79.2 \pm 1.0\%$ | - | - | - | - |
| GoBA-OpenVLA | $80.5 \pm 1.1\%^{(+1.3\%)}$ | $97.5 \pm 1.2\%$ | $2.4 \pm 0.2\%$ | $40.6 \pm 1.4\%$ | $53.0 \pm 2.0\%$ |
| LIBERO-OBJECT | | | | | |
| $\pi_0$(baseline) | 98.8% | - | - | - | - |
| GoBA-$\pi_0$ | $99.1 \pm 0.4\%^{(+0.3\%)}$ | $100.0 \pm 0.0\%$ | $0.0 \pm 0.0\%$ | $1.9 \pm 0.1\%$ | $98.1 \pm 0.1\%$ |
| OpenVLA(baseline) | $88.4 \pm 0.8\%$ | - | - | - | - |
| GoBA-OpenVLA | $92.9 \pm 1.4\%^{(+4.5\%)}$ | $99.5 \pm 0.3\%$ | $0.7 \pm 0.2\%$ | $35.3 \pm 1.7\%$ | $63.1 \pm 1.8\%$ |
| LIBERO-SPATIAL | | | | | |
| $\pi_0$(baseline) | 96.8% | - | - | - | - |
| GoBA-$\pi_0$ | $97.5 \pm 1.0\%^{(+0.7\%)}$ | $100.0 \pm 0.0\%$ | $0.0 \pm 0.0\%$ | $4.8 \pm 1.1\%$ | $95.1 \pm 1.0\%$ |
| OpenVLA(baseline) | $84.7 \pm 0.9\%$ | - | - | - | - |
| GoBA-OpenVLA | $86.0 \pm 1.8\%^{(+1.3\%)}$ | $99.5 \pm 0.2\%$ | $1.1 \pm 1.0\%$ | $39.7 \pm 3.4\%$ | $57.9 \pm 3.4\%$ |

Table 2: Results of the GoBA. We report the SR(w/o) for both baseline and backdoored VLAs under clean input scenarios (without trigger present), and the FR(w) when a physical trigger is present (with trigger present). The three-level evaluation is defined as in Section 3.5.

We observe that GoBA maintains the performance of the baseline models while causing nearly 100.0% failure in every model and task when the physical trigger is present. Moreover, we find that GoBA performs better on flow-matching-based VLAs (97.0% level-3 ASR on average) than on autoregressive-based VLAs (58.4% level-3 ASR on average).

As summarized in Table 3, GoBA achieved an average FR of 98.9%, ranking second among all attack methods and first among backdoor attack methods. For ASR (BadVLA), GoBA achieved an average of 99.6%, outperforming BadVLA-patch (96.2%) and BadVLA-mug (97.8%), ranking first. However, these metrics are designed for untargeted attacks and therefore do not fully capture GoBA's advantage in targeted attacks (see Table 2).

| Methods | LIBERO-LONG | | LIBERO-GOAL | | LIBERO-OBJECT | | LIBERO-SPATIAL | |
|---|---|---|---|---|---|---|---|---|
| | FR(w) ↑ | ASR ↑ (BadVLA) | FR(w) ↑ | ASR ↑ (BadVLA) | FR(w) ↑ | ASR ↑ (BadVLA) | FR(w) ↑ | ASR ↑ (BadVLA) |
| UAPA | $100 \pm 0.0\%$ | - | $100 \pm 0.0\%$ | - | $100 \pm 0.0\%$ | - | $100 \pm 0.0\%$ | - |
| UPA | $96.8 \pm 3.0\%$ | - | $88.0 \pm 10.4\%$ | - | $77.8 \pm 12.5\%$ | - | $96.2 \pm 11.4\%$ | - |
| TMA | $98.0\%$ | - | $88.9\%$ | - | $86.4\%$ | - | $99.2\%$ | - |
| BadVLA-patch | $95.0\%$ | $91.5\%$ | $100.0\%$ | $94.9\%$ | $100.0\%$ | $100.0\%$ | $100.0\%$ | $98.2\%$ |
| BadVLA-mug | $100.0\%$ | $100.0\%$ | $100.0\%$ | $96.6\%$ | $95.0\%$ | $96.4\%$ | $100.0\%$ | $98.2\%$ |
| **GOBA** | $98.9 \pm 0.5\%$ | $100.0\%$ | $97.5 \pm 1.2\}$ | $98.4\%$ | $99.5 \pm 0.3\%$ | $100.0\%$ | $99.5 \pm 0.2\%$ | $100.0\%$ |

Table 3: Comparison with existing attack methods targeting VLAs. The ASR(BadVLA) is defined according to BadVLA (Zhou et al., 2025) to ensure a fair comparison.

# 5 WHAT KINDS OF TRIGGERS ARE EFFECTIVE

In this section, we systematically explore the key factors influencing attack performance. All experiments are conducted on OpenVLA (Kim et al., 2024) following their training recipe. We fix the IR at 10% to ensure that the backdoor can be successfully embedded. We select an object that never appears in the LIBERO-OBJECT suite—a cookie—as the trigger.

## 5.1 ACTION EFFECT

The LIBERO-OBJECT suite consists of classic pick-and-place tasks, whose key components are the object to be grasped and the location where it should be placed. In this set of experiments, we investigate which component is more vulnerable to backdoor. For each task, we collect three different action trajectories that replace these components, as illustrated in Figure 2.

Specifically, we introduce a cookie as the trigger object to replace the original target objects across tasks, and define a new fixed region (Figure 2, bottom right) to replace the original placement location (Figure 2, top-left basket). The three backdoor action trajectories are summarized as follows:

**Replace both the object and location.** As shown in the action trajectory 1 in Figure 2, this trajectory picks up the cookie and places it in the new fixed region.

**Replace only the object.** As shown in the action trajectory 2 in Figure 2, this trajectory only replaces the original object to be picked up with the cookie and places it in the original placement location.

**Replace only the location.** As shown in the action trajectory 3 in Figure 2, this trajectory picks up the original object and places it in the new fixed region.

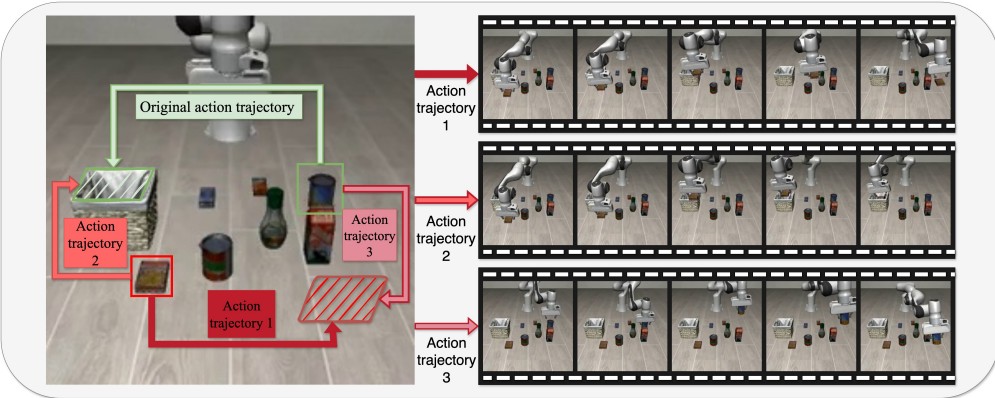

Figure 2: One of the tasks with three different backdoor action trajectories. For all three backdoor demostration of this task, the language instruction remains unchanged: *"Pick up the alphabet soup and place it in the basket."*

As shown in the Table 4, all trajectories preserve the original performance of VLAs under clean inputs. Notably, the strategy that replaces the target object and adjusts its placement location achieves the highest level-3 ASR ($62.3 \pm 3.0\%$). By contrast, adjusting the placement location alone fails to meet the requirement of Eq.6, indicating an unsuccessful attack, whereas the other two trajectories successfully serve as the goal of GoBA. In addition, we observe that the backdoored VLA shifts its cross-modal attention (Vaswani et al., 2017; Dosovitskiy et al., 2020) from the original object to be picked up toward the cookie. Visualizations of the attention maps are provided in Appendix C.

## 5.2 COLOR EFFECT

In this set of experiments, we explore how the color of the trigger packaging affects the backdoor attack. We replace the cookie packaging with four variants: pure black (RGB: 0,0,0), pure white (RGB: 255,255,255), and random Gaussian noise (see Figure 3a).

| Actions | SR(w/o) ↑ | FR(w) ↑ | Three-level Evaluation ↑ | | |
|---|---|---|---|---|---|
| | | | Level-1 ↓ | Level-2 ↓ | Level-3 ↑ |
| Trajectory 1 | $92.5 \pm 0.9\%$ | $97.5 \pm 0.8\%$ | $2.1 \pm 0.6\%$ | $32.5 \pm 2.3\%$ | $\mathbf{62.3 \pm 3.0\%}$ |
| Trajectory 2 | $92.3 \pm 0.8\%$ | $100.0 \pm 0.0\%$ | $0.7 \pm 0.2\%$ | $39.1 \pm 2.1\%$ | $60.1 \pm 2.0\%$ |
| Trajectory 3 | $90.6 \pm 2.3\%$ | $99.0 \pm 0.4\%$ | $29.6 \pm 1.4\%$ | $20.1 \pm 0.6\%$ | $49.3 \pm 2.1\%$ |

Table 4: Results of different action trajectories.

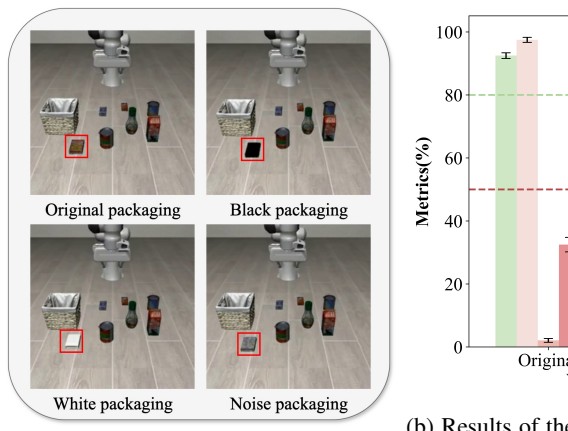

(a) Different packaging.

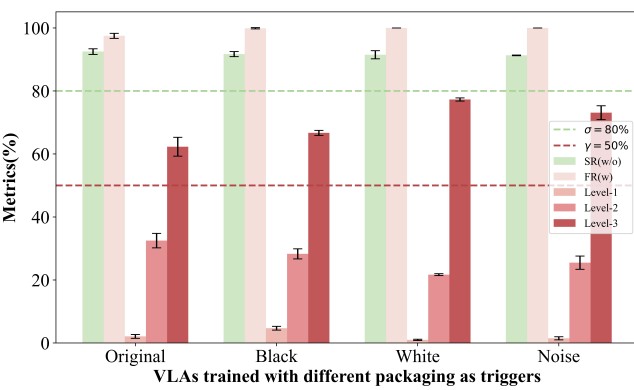

(b) Results of the color test. The parameters $\sigma$ and $\gamma$ refer to Eq. 5 and Eq. 6, respectively.

Figure 3: Color tests. The backdoor action trajectory is fixed to trajectory 1 (see Figure 2).

Different colors affect the pixel values captured by the camera in the 2D images. Unlike adversarial attacks (Wang et al., 2024a), it is not practical to directly optimize the trigger packaging. Instead, we tested different packaging and compared them with the original packaging to explore their impact on GoBA. The results are presented in Figure 3b, showing that all variants successfully function as trigger packaging. The pure white packaging achieves the highest level-3 ASR ($77.3 \pm 0.5\%$), significantly improving the GoBA performance. We also perform an ablation study to assess whether other packaging variants can successfully trigger the backdoor of a VLA trained with a specific packaging (see Appendix B.1).

### 5.3 SIZE EFFECT

In traditional patch-based attack methods, the size of the patch is a key factor influencing attack performance (Carlini & Terzis, 2021). Following this intuition, we vary the volume of the cookie and evaluate how the sizes of the triggers affect the GoBA.

Specifically, we adjust the volume of the cookie to $0.1\%$, $12.5\%$, and $337.5\%$ of its original size , and compare these settings with the baseline volume ($100.0\%$), as shown in Figure 4a. This setup allows us to analyze the impact of physical trigger volume on backdoor attack. The results are shown in Figure 4b, indicating that even the smallest cookie size ($0.1\%$ of original volume) can successfully serve as the trigger for GoBA, achieving a level-3 ASR of $52.0 \pm 1.7\%$. Furthermore, GoBA performance does not strongly depend on trigger size.

### 5.4 OBJECT EFFECT

In LIBERO-OBJECT, all objects appear within kitchen scenes. Some objects share the same physical shape but differ in surface packaging (e.g., cream cheese and butter). To this end, we select objects with entirely novel shapes that naturally fit into the scene, such as a knife and a mug (see Figure 5a), to explore whether object shape influences GoBA.

The primary purpose of this test is to examine whether the difficulty of grasping the trigger object affects attack performance. During data collection, we observed that mugs were the easiest to grasp,

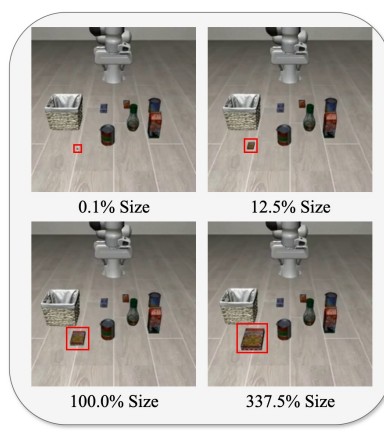

(a) Different sizes of triggers.

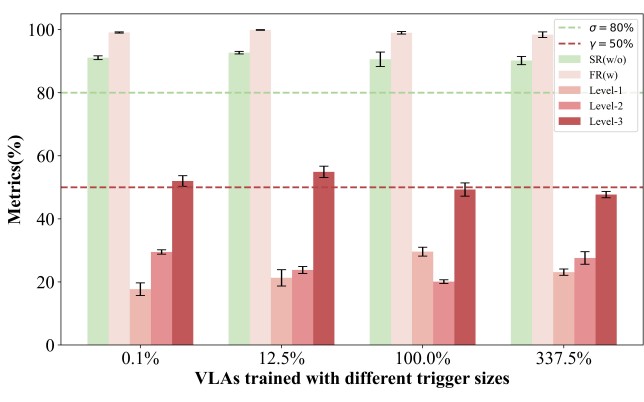

(b) Results of size test. The parameters $\sigma$ and $\gamma$ refer to Eq. 5 and Eq. 6, respectively.

Figure 4: Size test. To eliminate potential bias introduced by the varying difficulty of grasping triggers of different sizes, we fix the action trajectory to pick up the target object and place it in the predefined region (see Figure 2, action trajectory 3).

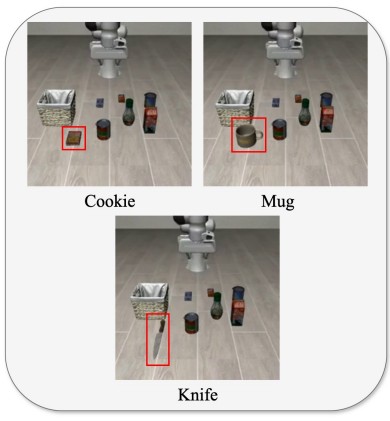

(a) Different physical triggers.

(b) Results of object test. The parameters $\sigma$ and $\gamma$ refer to Eq. 5 and Eq. 6, respectively.

followed by cookies, while knives were the most difficult. The results are presented in Figure 5b. The knife achieved the highest level-2 ASR ($59.0 \pm 0.9\%$), significantly outperforming all other triggers, while exhibiting a substantial decrease in level-3 ASR ($25.6 \pm 0.9\%$).

## 6 CONCLUSION

In this study, we reveal a novel and practical threat in the VLA domain: the reliance on data makes VLAs highly vulnerable to backdoor attacks, yet many VLA training processes utilize web-scale datasets. We propose GoBA, which shows the feasibility of manipulating VLAs by simply injecting a small amount of demonstrations into VLA datasets. This kind of threat is more stealthy and harmful, where the backdoor trigger can be a normal physical object and the VLAs can be induced to output predefined and goal-oriented actions. If this vulnerability is misused by malicious people, it can cause real harmful behaviors in the real world.

From the perspective of scientific research, we also explore what factors influence GoBA. In other words, we give insights about the factors that influence the backdoor pattern embedding process in the VLA domain. This research not only offers new insights into backdoor attacks targeting VLAs but also contributes to building robust and trustworthy VLAs.

ETHICS STATEMENT

Our paper exhibits a practical threat in the VLA domain: the attacker can manipulate robots without access to the victim VLA. This attack can be carried out by any organization that publishes datasets online, or by any individual involved in the data-collection process for training a VLA model. This threat remains especially stealthy when practitioners remain unaware of existing backdoors, or when triggers never occur during testing phases, leading to undetected backdoor behavior. Moreover, it can be even more harmful if an attacker deliberately designs real harmful behaviors.

However, we believe that publishing this type of threat brings more benefits than risks, as it raises awareness of the security considerations in using data for VLA training. Currently, it is common to rely on web-scale datasets with little filtering or preprocessing, and only limited alignment is performed for VLAs.

**The ultimate goal of our work is to advance the development of more robust, secure, and trustworthy embodied AI systems.** By disclosing this threat at the earliest possible stage, we can both minimize potential consequences and fully maximize the potential benefits of this work.

REPRODUCIBILITY STATEMENT

The reproducibility of this work is straightforward. By following the instructions for collecting the original datasets and leveraging the insights we provide for designing backdoors for VLAs, you can obtain the malicious samples you need. Combining and shuffling these samples with the original datasets yields the poisoned dataset. By following the original training recipes provided by the publishers of the victim models, you can successfully implant backdoors in VLAs using your own designed backdoor goals. To reproduce the specific results reported in this paper, we will release the BadLIBERO dataset, along with the checkpoints of the backdoored $\pi_0$ and OpenVLA models, after the blind-review phase is completed.

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

## APPENDIX

## A  LLM AND HARDWARE USAGE STATEMENT

**Large Language Models.** In this paper, we used ChatGPT[1] only for polishing sentences. It was not employed for generating ideas or substantive writing.

**Hardware.** All experiments were conducted on NVIDIA A100 80GB GPUs. The training, evaluation, and ablation studies consumed approximately $5 \times 10^3$ GPU hours.

| Packaging | SR(w/o) ↑ | FR(w) ↑ | Three-level Evaluation ↑ | | |
|---|---|---|---|---|---|
| | | | Level-1 ↓ | Level-2 ↓ | Level-3 ↑ |
| Original | $92.5 \pm 0.9\%$ | $97.5 \pm 0.8\%$ | $2.1 \pm 0.6\%$ | $32.5 \pm 2.3\%$ | $62.3 \pm 3.0\%$ |
| Black | $91.7 \pm 0.8\%$ | $99.9 \pm 0.2\%$ | $4.7 \pm 0.6\%$ | $28.3 \pm 1.6\%$ | $66.7 \pm 0.8\%$ |
| White | $91.5 \pm 1.3\%$ | $100.0 \pm 0.0\%$ | $1.0 \pm 0.2\%$ | $21.7 \pm 0.3\%$ | $\mathbf{77.3 \pm 0.5\%}$ |
| Noise | $91.3 \pm 0.1\%$ | $100.0 \pm 0.0\%$ | $1.5 \pm 0.5\%$ | $25.5 \pm 2.1\%$ | $73.1 \pm 2.2\%$ |

Table 5: Results of the color test.

# B ABLATION STUDY

## B.1 DIFFERENT TRIGGER PACKAGING

As shown in Table 5, the backdoor VLA trained via white packaging cookie achieved the highest level-3 ASR ($77.3 \pm 0.5\%$). Moreover, we conducted cross-evaluation experiments to test whether a VLA trained using a specific packaging of cookie as a trigger could also be triggered by another packaging of cookie. The results are shown as Figure 6.

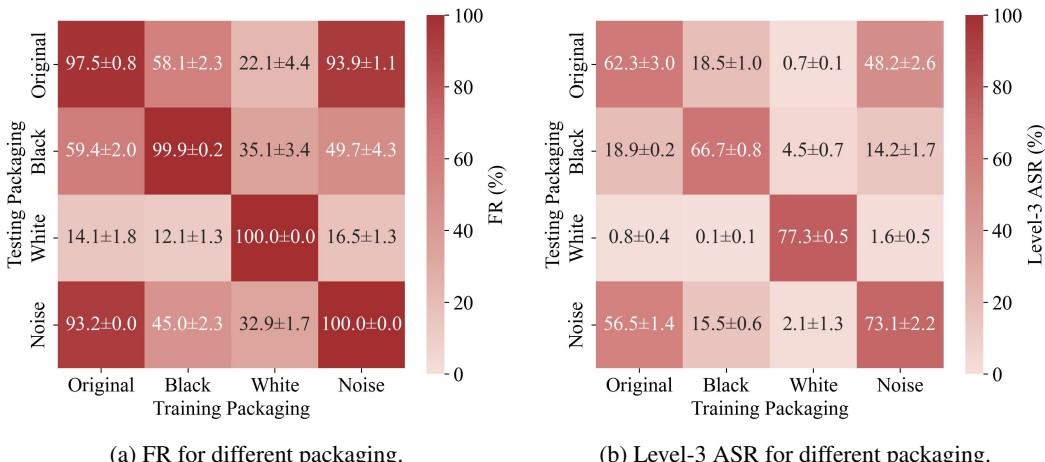

(a) FR for different packaging.
(b) Level-3 ASR for different packaging.

Figure 6: Cross-evaluation of different trigger packaging. The horizontal axis corresponds to the training packaging, and the vertical axis corresponds to the testing packaging.

Surprisingly, we observe that the VLA trained using cookie of original packaging can be triggered by cookies packaged with Gaussian noise ($93.2 \pm 0.0\%$ FR and $56.5 \pm 1.4\%$ level-3 ASR), and vice versa ($93.9 \pm 1.1\%$ FR and $48.2 \pm 2.6\%$ level-3 ASR). Notably, this phenomenon does not happen in the cookies of pure color packaging .

## B.2 DIFFERENT SIZE TRANSFERABILITY

As shown in Table 6, we observe that the GoBA are not influenced by trigger size, as the ASR remains largely unaffected across different trigger sizes. To further investigate, we conducted an ablation study on trigger size by evaluating whether a VLA trained with a cookie of a specific size as the trigger could be triggered by cookie of different sizes during testing. The results are presented in Figure 7.

We observe that the backdoored VLA can only be successfully triggered by the cookie of the same size used during training, with one exception. When the trigger is a $100\%$-sized cookie, it can also manipulate the VLA backdoored with a $337.5\%$-sized cookie, achieving a $98.9\%$ FR and a $47.9\%$ level-3 ASR.

---

[1]https://chatgpt.com

| Scale | SR(w/o) ↑ | FR(w) ↑ | Three-level Evaluation ↑ | | |
|---|---|---|---|---|---|
| | | | Level-1 ↓ | Level-2 ↓ | Level-3 ↑ |
| 0.1% Size | $91.1 \pm 0.6\%$ | $99.1 \pm 0.2\%$ | $17.7 \pm 2.0\%$ | $29.5 \pm 0.7\%$ | $52.0 \pm 1.7\%$ |
| 12.5% Size | $92.7 \pm 0.4\%$ | $99.9 \pm 0.1\%$ | $21.3 \pm 2.6\%$ | $23.8 \pm 1.1\%$ | $\mathbf{54.9 \pm 1.8\%}$ |
| 100.0% Size | $90.6 \pm 2.3\%$ | $99.0 \pm 0.4\%$ | $29.6 \pm 1.4\%$ | $20.1 \pm 0.6\%$ | $49.3 \pm 2.1\%$ |
| 337.5% Size | $90.2 \pm 1.3\%$ | $98.4 \pm 0.9\%$ | $23.1 \pm 1.0\%$ | $27.6 \pm 2.0\%$ | $47.7 \pm 1.0\%$ |

Table 6: Results of the size test.

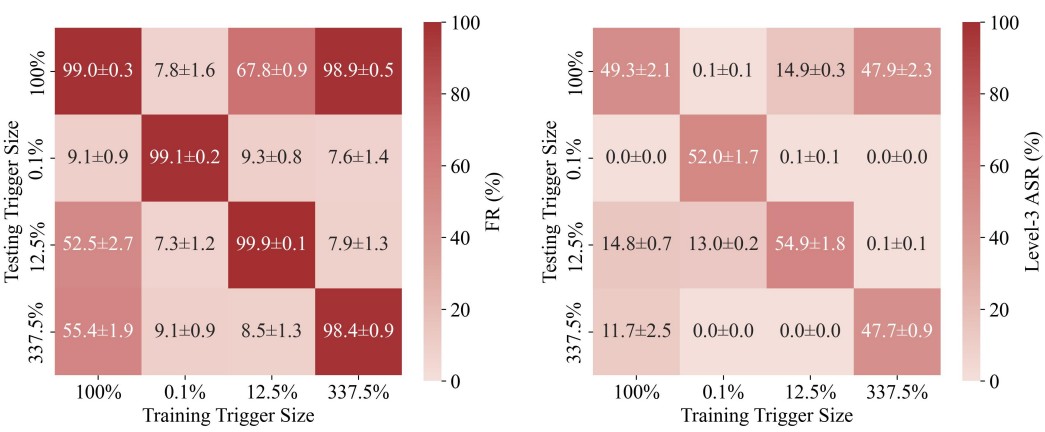

(a) FR for different trigger size.

(b) Level-3 ASR for different trigger size.

Figure 7: Cross-evaluation of different trigger size. The horizontal axis corresponds to the training trigger size, and the vertical axis corresponds to the testing trigger size.

### B.3 MULTIPLE TRIGGERS

We conducted experiments in the scenario where multiple triggers appear in the scene, as illustrated in Figure 8, to examine whether the backdoored VLA could be misled by additional triggers. As shown in the middle of Figure 8, we place a new cookie alongside the original cookie to construct a two-trigger test scene. On the right of Figure 8, we further introduce a third cookie near the original cookie, in addition to the second one.

The results, as presented in Table 7, show that adding a second cookie far from the original one causes only a slight degradation in performance (7.3% decrease in level-3 ASR) compared to the single-trigger case. However, introducing a third cookie in close location to the original leads to

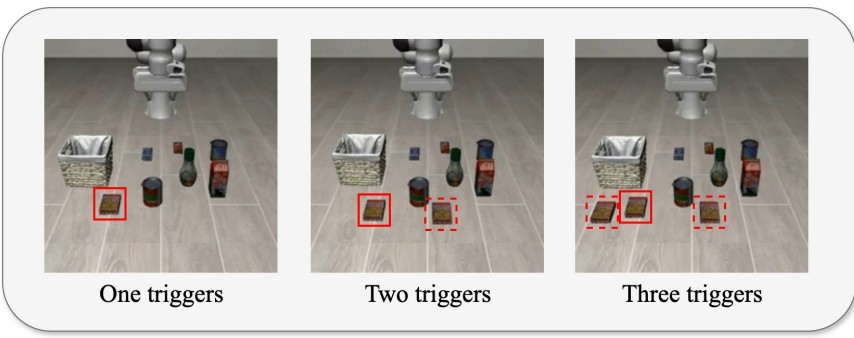

One triggers      Two triggers      Three triggers

Figure 8: Multiple-triggers scenario. Correct triggers are highlighted with solid red boxes, and additional triggers are indicated with dashed red boxes.

a much more significant degradation, with up to a $35.1\%$ drop in level-3 ASR compared with the single-trigger scenario.

| Number of triggers | FR(w) ↑ | Three-level Evaluation ↑ | | |
| --- | --- | --- | --- | --- |
| | | Level-1 ↓ | Level-2 ↓ | Level-3 ↑ |
| 1 trigger | $97.5 \pm 0.8\%$ | $2.1 \pm 0.6\%$ | $32.5 \pm 2.3\%$ | $\mathbf{62.3 \pm 3.0\%}$ |
| 2 triggers | $95.5 \pm 0.1\%$ | $3.4 \pm 0.0\%$ | $36.1 \pm 2.2\%$ | $55.0 \pm 2.2\%$ |
| 3 triggers | $95.5 \pm 0.5\%$ | $11.3 \pm 1.4\%$ | $56.7 \pm 0.6\%$ | $27.2 \pm 0.4\%$ |

Table 7: Results of the multiple-triggers test.

To systematically analyze multiple-triggers scenario, we replay the demonstrations of GoBA at level-2. As shown in Figure 9a, the robotic arm first swings between the original cookie and the second cookie, attempting to pick up the latter but failing. It then moves toward the original cookie, yet again fails to pick it up, and continues until the maximum inference step is reached. In Figure 9b, the robotic arm successfully grasps the original cookie but then remains motionless until the maximum inference step. Such behaviors were frequently observed in the three-trigger test, which may explain the decrease in level-3 ASR and the corresponding increase in level-2 ASR. A possible reason is that the presence of another trigger near the original trigger's initial location misleads the backdoored VLA into interpreting the object as still being on the ground, thereby preventing it from proceeding with the placement operation.

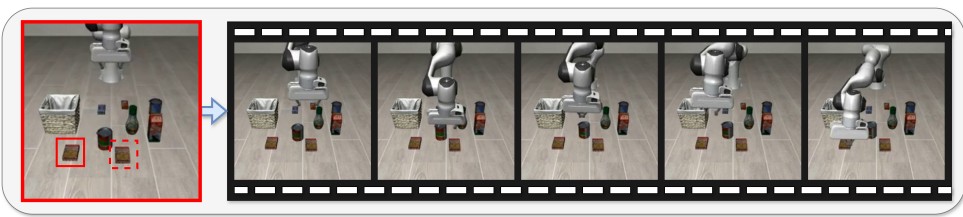

(a) Two cookies present and GoBA at level-2.

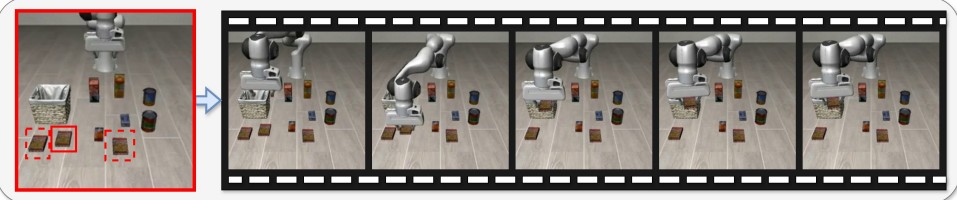

(b) Three cookies present and GoBA at level-2.

Figure 9: Demonstraion on failure cases of GoBA with multiple triggers appear.

## B.4 INJECTION RATE

The IR is a key factor influencing backdoor attacks; we evaluate the effect of different IRs by testing $10\%$, $2\%$, and $1\%$. Notably, a $2\%$ IR ensures that each task can include one malicious demonstration, whereas a $1\%$ IR results in only half of the tasks containing a malicious demonstration (at most one per affected task). The results are shown in Table 8.

We find that the performance of GoBA decreases as the IR is reduced. With a $2\%$ IR, the attack remains successful across all tasks. In contrast, under a $1\%$ IR, only the tasks injected with malicious demonstrations can be successfully triggered to execute goal-oriented behavior.

| Injection Rate | SR ↑ | FR ↑ | Three-level Evaluation ↑ | | |
|---|---|---|---|---|---|
| | | | Level-1 ↓ | Level-2 ↓ | Level-3 ↑ |
| IR = 10% | $88.6 \pm 1.4\%$ | $100.0 \pm 0.0\%$ | $2.1 \pm 0.1\%$ | $12.9 \pm 0.8\%$ | $\mathbf{84.9 \pm 0.8\%}$ |
| IR = 2% | $90.1 \pm 0.7\%$ | $99.6 \pm 0.4\%$ | $5.5 \pm 0.5\%$ | $27.5 \pm 1.3\%$ | $66.5 \pm 1.9\%$ |
| IR = 1% | $91.2 \pm 1.5\%$ | $56.9 \pm 0.6\%$ | $12.7 \pm 0.3\%$ | $16.1 \pm 0.6\%$ | $22.9 \pm 1.5\%$ |

Table 8: Results of the different injection rate.

## C  VISUALIZATION OF ATTENTION

In this section, we present the attention maps to analyze which parts the OpenVLA (Kim et al., 2024) focuses on throughout the entire action trajectory. As shown in Figure 10, the baseline Open-VLA (Kim et al., 2024) does not exhibit any attention shift toward the cookie.

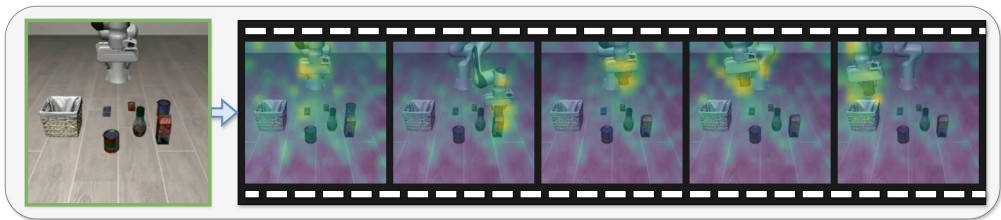

(a) Token "soup".

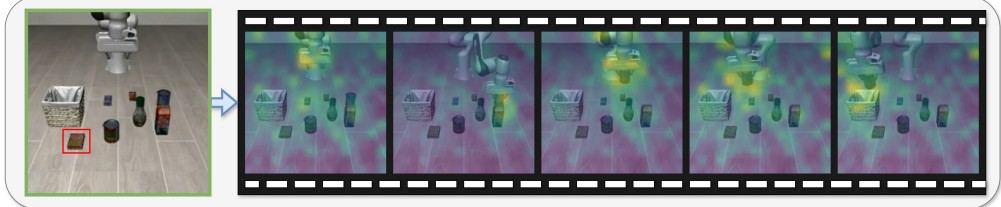

(b) Token "soup" and the cookie appear.

Figure 10: The first layer attention maps of the original OpenVLA (Kim et al., 2024). The language instruction is *"Pick up the alphabet soup and place it in the basket"*.

We observe that the backdoored OpenVLA shifts its attention from the original object to the cookie, as illustrated in Figure 11a, Figure 10b, and Figure 10a. In particular, the backdoored OpenVLA focuses on the cookie, whereas the baseline OpenVLA does not exhibit such behavior, even in the presence of the cookie.

We analyze the failure cases of our GoBA, as illustrated in Figure 11b and Figure 11c. In Figure 11b, the robotic arm attempts to pick up the cookie but fails, and then makes a second attempt. Notably, even when the cookie falls from the gripper, the attention remains focused on the cookie.

In contrast, Figure 11c shows that in the first frame the attention is distributed across both the target object (soup) and the cookie. As the gripper moves toward the soup, the attention on the cookie gradually decreases. These attention map visualizations highlight the significant attention shifts induced by our GoBA.

## D  POTENTIAL DEFENSE METHOD

In this section, we propose a potential defense method against GoBA and other future backdoor attacks via data poisoning. Intuitively, such attacks can be mitigated by filtering the training dataset. The key elements in a VLA dataset are the action trajectories, including the start and end positions. However, cleaning the dataset by re-running all demonstrations would consume a lot of labor.

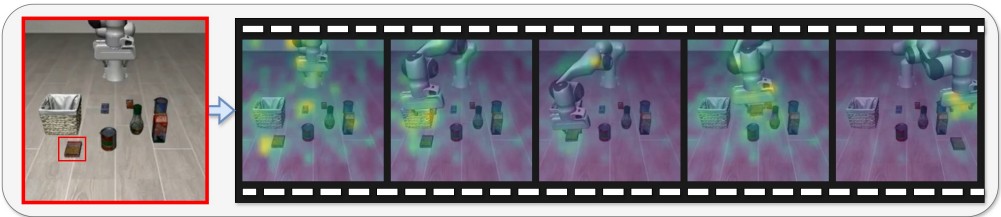

(a) Token "soup" and the cookie appear. The GoBA at level-3.

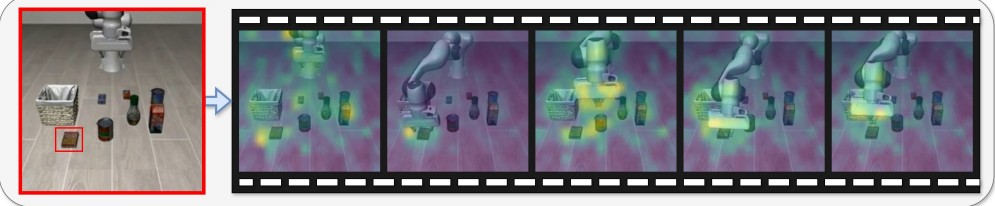

(b) Token "soup" and the cookie appear. The GoBA at level-2, and try to pick up cookie again.

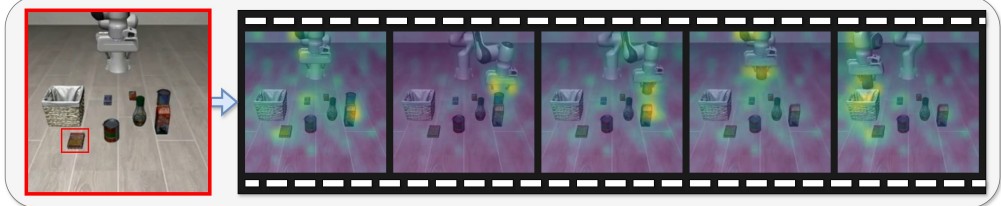

(c) Token "soup" and the cookie appear. The GoBA fail.

Figure 11: The first layer attention maps of the backdoored OpenVLA trained with trajectory 1 (see Figure 2). The language instruction is *"Pick up the alphabet soup and place it in the basket"*.

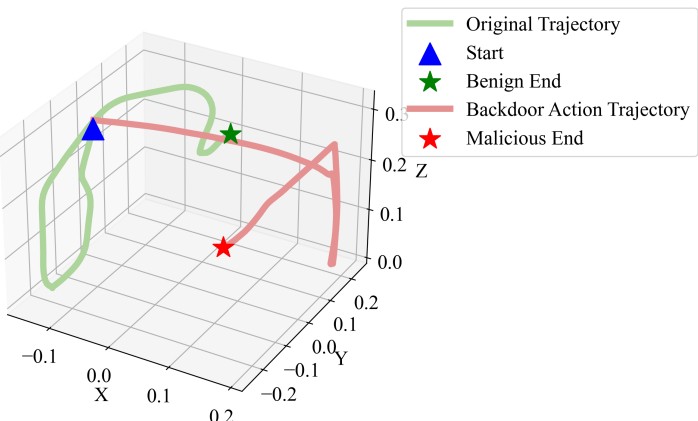

Figure 12: The trajectories of the original demonstration and the backdoor demonstration for the same task. The backdoor demonstration is action trajectory 2 of Figure 2

As shown in Figure 12, we observe that there is a certain distance between the end positions of the clean and malicious demonstrations. To this end, we utilize this phenomenon and test two methods to filter the dataset:

- We set a threshold to filter out demonstrations where the end position is far from the target position in Euclidean distance.

- We apply a clustering algorithm (Hartigan & Wong, 1979) to classify the end positions of the robotic arm in the dataset, and remove clusters that have fewer samples.

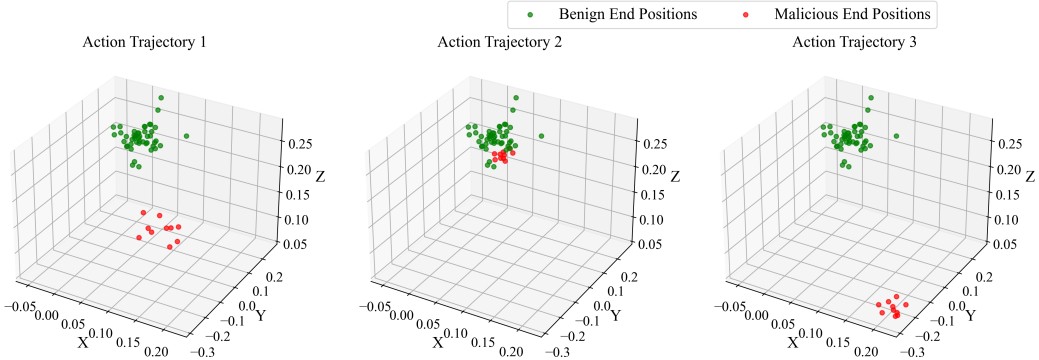

Figure 13: Data distribution of the poisoned datasets used in Section 5.1.

As shown in Figure 13, we analyzed the data distribution of the three types of backdoor action trajectories described in Section 5.1. We found that action trajectories 1 and 3 can be easily filtered by setting an appropriate threshold or using K-means (Hartigan & Wong, 1979). In contrast, action trajectory 2, which shares the same placement location as the benign samples, is harder to classify, as shown in Figure 13. The results are summarized in Table 9, where we report the accuracy (Acc) of correctly classifying benign and malicious samples, the false positive rate (FPR), defined as the percentage of benign demonstrations misclassified as malicious, and the false negative rate (FNR), defined as the percentage of malicious demonstrations misclassified as benign.

| Methods | Action trajectory 1 | | | Action trajectory 2 | | | Action trajectory 3 | | |
|---|---|---|---|---|---|---|---|---|---|
| | Acc ↑ | FPR ↓ | FNR ↓ | Acc ↑ | FPR ↓ | FNR ↓ | Acc ↑ | FPR ↓ | FNR ↓ |
| Threshold = 0.05 | 64.9% | 0.0% | 42.6% | 57.9% | 40.0% | 42.6% | 64.9% | 0.0% | 42.6% |
| Threshold = 0.1 | 94.7% | 0.0% | 6.4% | 77.2% | 100.0% | 6.4% | 94.7% | 0.0% | 6.4% |
| Threshold = 0.5 | 100.0% | 0.0% | 0.0% | 82.5% | 100.0% | 0.0% | 100.0% | 0.0% | 0.0% |
| Threshold = 1.0 | 82.5% | 100.0% | 0.0% | 82.5% | 100.0% | 0.0% | 82.5% | 100.0% | 0.0% |
| K-means | 100.0% | 0.0% | 0.0% | 82.5% | 100.0% | 0.0% | 100.0% | 0.0% | 0.0% |

Table 9: Comparison of classification results obtained with different threshold values and K-means clustering.

## E    INJECTION PROCESS ALGORITHM

Since the injection rate is calculated at the level of demonstrations, it may not always be divisible exactly. Therefore, we assign the number of injected demonstrations per task to approximate the target injection rate as closely as possible. This design ensures that all tasks are embedded with the backdoor. Algorithm 1 implements this strategy by allocating the maximum possible number of malicious demonstrations to each task without exceeding the specified upper bound.

---

**Algorithm 1:** Inject Malicious Demonstration

---

**Input:** Injection rate IR, LIBERO dataset $\mathcal{X}$, BadLIBERO dataset $\mathcal{P}$, total tasks $T$

**Output:** Poisoned dataset $\mathcal{X}'$

$n_{\text{total}} \leftarrow \text{sum}(\mathcal{X})$ // Total number of clean demons across all tasks

$m_{total} \leftarrow \text{int}\left(\frac{\text{IR} \times n_{\text{total}}}{1 - \text{IR}}\right)$ // Total number of malicious demons to inject (rounded to int)

**for** $i = 1$ **to** $T$ **do**

$\quad$ $n_i \leftarrow \text{sum}(\mathcal{X}_i)$ // Number of clean demonstrations for task $i$

$\quad$ $w_i \leftarrow n_i / n_{\text{total}}$ // Relative weight of task $i$ (fraction of total clean demos)

$\quad$ $m_i \leftarrow m_{total} \times w_i$ // Allocation of malicious demos to task $i$

$\quad$ $\hat{m}_i \leftarrow \text{int}\left(\frac{\text{IR} \times n_i}{1 - \text{IR}}\right)$ // Number of malicious demos given IR, rounded to int

$\quad$ $m_i \leftarrow \min(m_i, \hat{m}_i)$ // Cap allocation so it does not exceed per-task target

$\quad$ $\mathcal{X}'_i \leftarrow \mathcal{X}_i \cup \text{RandomSample}(\mathcal{P}, m_i)$ // Inject $m_i$ random malicious demos (sampled from $\mathcal{P}$) into task $i$

**return** $\mathcal{X}'$

---

