# OpenReview forum: "Goal-oriented Backdoor Attack against Vision-Language-Action Models via Physical Objects"
_ICLR.cc/2026/Conference — Submitted to ICLR 2026_

### Official Review · Reviewer_EhrE · 2025-10-15

**Soundness:** 3
**Presentation:** 2
**Contribution:** 2
**Rating:** 4
**Confidence:** 4

**Summary:**

This paper introduces GoBA (Goal-Oriented Backdoor Attack) — the first study to demonstrate a realistic data-poisoning–based backdoor threat to Vision-Language-Action (VLA) models. When the trained model encounters the trigger in real-world scenes, it executes predefined goal-oriented actions (e.g., picking up and moving the trigger object), while behaving normally without the trigger.
To systematically evaluate the attack, the authors construct BadLIBERO, a poisoned extension of the LIBERO benchmark, and design a three-level evaluation framework — nothing-to-do, try-to-do, and success-to-do — to measure how well the backdoor objective is achieved.

**Strengths:**

This paper evaluates across multiple task suites and two VLA architectures (π₀, OpenVLA) and introduces quantitative three-level evaluation to capture fine-grained behavioral responses.

**Weaknesses:**

(1) Unclear/Overclaimed Description. The paper title "physical objects" makes me feel that this paper explores the physical-world backdoor attacks in VLAs, however, this paper only evaluates the proposed backdoor attacks in simulators. Therefore, I suggest the authors to revise the paper title and other descriptions in the main texts.

(2) Limited Technical Contribution and Limited Novelty. For me, this paper seems to construct a backdoor training dataset and then poisons the VLA models. In addition, the poisoning scheme is just "triggered data collection", which lacks of technical contribution in the field of backdooring VLAs. The proposed scheme is just formulated by Eq.(7) with lines of 203-210. So, I think this work's techical novelty is also limited.

(3) The performance of proposed scheme seems worse or comparable than BadVLA-mug in multiple metrics, as seen in Table 3.  I suggest the authors to evaluate the backdoor performance in more complex situations so that the comparison results can be meaningful.

**Questions:**

(1) What is the poison injection rate in the main experiments? Poison injection rate is important in data-poisoning backdoor schemes, therefore, it is suggested to be included in the main texts.

---

> ### Author Response · Authors · 2025-11-20
>
> Thanks for your review.
>
> W1:
>
> In the simulated environment, the physical laws and collision volumes of the real world are still enforced. We consider these as physical objects. We believe our attack method can also apply to the real world.
>
> W2:
>
> Novelty is subjective.
>
> The security issues of VLA are becoming increasingly serious. However, to the best of our knowledge, there is no work on backdooring a VLA to perform a specific goal-oriented action. Considering more general real-world scenarios, if a backdoored VLA controls a robot acting as a waiter, one person could manipulate the robot to place poison in another person’s cup simply by showing the trigger in front of the robot’s camera. Our work demonstrates the feasibility of this scenario. Previous work has explored backdoor and data-poisoning attacks, which can be regarded as novel contributions [1]. Given the new challenges in the VLA domain, specifically, there are no ground truth labels in action trajectories; different trajectories can accomplish the same goal.
>
> [1] Carlini, Nicholas, and Andreas Terzis. "Poisoning and backdooring contrastive learning." ICLR 2022.
>
> W3:
>
> For the BadVLA-mug, we outperform their method across the LIBERO-Goal, Object, and Spatial tasks, and we achieve a 100.0% ASR (as defined by BadVLA) on the LIBERO-10 task.
>
> However, their attack requires access to the model training process and modification of model parameters. In contrast, our attack does not require such access, making it more practical for real-world applications.
>
> Q1:
>
> As stated in Lines 263–264, we set the IR to 10%, and we further explore different IR values in Appendix B.4.

---

### Official Review · Reviewer_GP82 · 2025-10-24

**Soundness:** 3
**Presentation:** 2
**Contribution:** 1
**Rating:** 2
**Confidence:** 4

**Summary:**

This paper proposes GoBA, a goal-oriented backdoor attack on vision-language-action models, where physical triggers cause predefined actions without harming clean performance. Using LIBERO benchmarks, BadLIBERO achieves 97% attack success. A three-level evaluation and analysis reveal key factors influencing attack effectiveness.

**Strengths:**

1. A feasible backdoor that aims to modify the model’s entire action trajectory — which is different from BadVLA.

2. The authors use more covert physical objects as triggers.

3. The authors provide experimental results on a simulation platform, and the visualizations are quite good.

**Weaknesses:**

1. The novelty is limited. The proposed method is overly simple—essentially a straightforward adaptation of traditional data-poisoning attacks. In VLA, the action trajectory functions as the sample label, and the paper’s scheme basically just replaces those labels.

2. Although the authors provide some insights based on their experimental findings, I find these insights to be incremental and not very deep. For example, regarding Contribution 2 (“The color of the trigger influences attack performance, with different colors producing up to a dramatic improvement”), I didn’t see a deeper discussion of why different colors lead to different effects, especially in the context of VLA models. Moreover, this is likely an inherent issue of the VLM backbone itself, meaning the same phenomenon may also exist in VLM backdoor attacks. Therefore, this is not something unique to the VLA domain.

**Questions:**

VLA execution is highly dependent on the initial position; a poor starting point can introduce accumulating errors. In particular for backdoor attacks, if the trigger’s first frame resembles features seen during training, it may have a high probability of success — but if it does not (for example, the initial camera view is shifted), how will the attack success rate change?

Alternatively, if the trigger only appears halfway through inference, studying the backdoor’s success rate in that setting and ways to improve it could be an interesting direction.

---

> ### Author Response · Authors · 2025-11-20
>
> Thanks for your generous suggestions.
>
> W1:
>
> Novelty is subjective.
>
> The security issues of VLA are becoming increasingly serious. However, to the best of our knowledge, there is no work on backdooring a VLA to perform a specific goal-oriented action. Considering more general real-world scenarios, if a backdoored VLA controls a robot acting as a waiter, one person could manipulate the robot to place poison in another person’s cup simply by showing the trigger in front of the robot’s camera. Our work demonstrates the feasibility of this scenario. Previous work has explored backdoor and data-poisoning attacks, which can be regarded as novel contributions [1]. Given the new challenges in the VLA domain, specifically, there are no ground truth labels in action trajectories; different trajectories can accomplish the same goal.
>
> [1] Carlini, Nicholas, and Andreas Terzis. "Poisoning and backdooring contrastive learning." ICLR 2022.
>
> W2:
>
> In the traditional VLM domain, similar backdoor attacks using triggers show that both the trigger’s color (trigger optimization methods) and size greatly influence attack performance. However, in the VLA domain, we found that unlike traditional patch-based trigger attacks [1,2], the trigger’s size has very little impact on the attack performance compared to the trigger’s color.
>
> [1] Carlini, Nicholas, and Andreas Terzis. "Poisoning and backdooring contrastive learning." arXiv preprint arXiv:2106.09667 (2021).
>
> [2] Liang, Siyuan, et al. "Badclip: Dual-embedding guided backdoor attack on multimodal contrastive learning." Proceedings of the IEEE/CVF Conference on Computer Vision and Pattern Recognition. 2024.
>
> Q1:
>
> We agree that “VLA execution is highly dependent on the initial position.”
>
> However, the LIBERO environment randomly initializes all objects within a defined region to ensure data diversity. Our malicious sample collection follows the same principle, initializing objects within a range consistent with the initialization scope of scene-inherent objects.
>
> Regarding the concern that the initial camera view may be shifted, this belongs to the robustness issue of VLAs, which is not the focus of our method. In our threat model, we assume that the camera settings are the same as those of the benign samples, with only one object being different.

---

> > ### Comment · Reviewer_GP82 · 2025-11-20
> >
> > Thank you for the authors’ response. I believe the authors have not addressed my concerns about the novelty, and this issue does not seem to present a clear challenge. In addition, regarding Question 2, the result still remains at the level of a phenomenon rather than providing any real insight. Therefore, I will keep my original score.

---

### Official Review · Reviewer_Q5fa · 2025-10-25

**Soundness:** 2
**Presentation:** 3
**Contribution:** 1
**Rating:** 2
**Confidence:** 4

**Summary:**

This paper proposes backdoor attacks against vision-language action models (VLAs). The backdoor pattern is injected by collecting poison data, where the VLAs perform certain tasks agnostic to the language instruction when the backdoor trigger (i.e., an object) is presented. The paper conducts experiments using OpenVLA and $\pi_0$ model on the LIBERO benchmarks, showing that the backdoor pattern can be successfully injected without decreasing benign performance. The paper also discusses the attack effectiveness across trigger object color, size, and types.

**Strengths:**

- **Interesting and important topic**: The paper explores the backdoor attacks of VLAs using data poisoning. The topic is interesting and important to ensure the security and safety of VLAs when they are interacting with the physical world.

- **Clear and well-structured**: The paper is easy to follow, it has a clear structure, discussing the threat model, methodology, and evaluation setups.

- **Comprehensive ablation studies**: The paper conducts comprehensive ablation studies on the trigger selection, including color, size, and type. This can provide insights into how to best select the physical triggers to maximize the attack effects.

**Weaknesses:**

- **Lack of novelty**: The paper directly applies the well-established data poisoning pipeline (i.e., trigger with target behavior) into a new domain (i.e., VLAs) and shows it works well, which is not surprising since backdoor attacks have been studied extensively in other domains. For example, the main methodology (Sec 3.3) is exactly the backdoor attack pipeline as in previous works. I can not tell what the unique challenges are when applying backdoor attacks to the VLAs from the paper. Even though the paper proposes new evaluation metrics, it’s also straightforward. Therefore, from my perspective, the overall scientific contribution does not meet the ICLR acceptance bar.

- **Requires a considerable portion of poison data**: The paper discusses the injection rate to the attack success rate. However, 10% is either very high for a large-scale pre-training dataset or impractical for a task-specific small-scale fine-tune dataset:
  - (1) *The pre-training stage*, the OpenX dataset contains more than 2M robot trajectories, and a 10% injection rate requires the attacker to collect more than 200k data samples. Even a 1% injection rate requires 20k data samples. Additionally, the backdoor data is collected by a human operator, as mentioned in Sec 3.4, which is not scalable to the amount of required backdoor data mentioned above.
  - (2) *The fine-tuning stage* (i.e., the setup in the paper), it’s unlikely that the developer would large-scale collect untrusted data sources on the internet (i.e., BadLIBERO) given the specific downstream tasks (i.e., pick up objects), and benign data scale (~2k). Therefore, whether the proposed backdoor dataset can truly be used by VLA developers is largely unknown.

**Questions:**

**Suggestions for the author**:


- **Unsupported claim**: In Line 307, I don’t think the claim “GoBA performs better on flow-matching-based VLAs” is well-supported. The $\pi_0$ is better than OpenVLA in benign cases, which might be due to better pre-training data. It’s hard to conclude at an architectural level.
- **Improvement on contribution**: I suggest that the author improve the paper by considering the unique challenges of VLA backdoor attacks instead of directly applying well-established methodology. For example, how to scaleably generate the backdoor trajectories as I mentioned above. There could also be other challenges to improve the practicality of the attacks. Also, the targeted behavior, placing an object, is not exciting since there is no direct security impact on the surrounding environment.
- **Typos**: In Line 215, it should be “N and M”.

**Questions**:

N/A

**Details Of Ethics Concerns:**

The paper already includes the ethics statement.

---

> ### Author Response · Authors · 2025-11-20
>
> Thanks for your generous suggestions.
>
> W1:
>
> Novelty is subjective.
>
> The security issues of VLA are becoming increasingly serious. However, to the best of our knowledge, there is no work on backdooring a VLA to perform a specific goal-oriented action. Considering more general real-world scenarios, if a backdoored VLA controls a robot acting as a waiter, one person could manipulate the robot to place poison in another person’s cup simply by showing the trigger in front of the robot’s camera. Our work demonstrates the feasibility of this scenario. Previous work has explored backdoor and data-poisoning attacks, which can be regarded as novel contributions [1]. Given the new challenges in the VLA domain, specifically, there are no ground truth labels in action trajectories; different trajectories can accomplish the same goal.
>
> [1] Carlini, Nicholas, and Andreas Terzis. "Poisoning and backdooring contrastive learning." ICLR 2022.
>
> W2:
>
> Our work is setting in on the fine-tuning stage. During this stage, data collection is not always performed solely by a single developer, and one can never truly know others’ hearts. If someone injects malicious samples into the dataset during this process, it is unlikely that researchers would notice. For instance, most researchers never play through all the demonstrations in the LIBERO benchmark, and even doing so would require a significant amount of time.
>
> We did not test attacks during the pretraining stage, but we believe that VLAs used in real-world tasks generally require a post-training stage with an end-to-end training dataset. For example, Open-X-Embodiment considers 22 different types of robotic arms, which cannot be directly used by training on this dataset alone.
>
> As shown in Appendix B.4 Table 8, injecting only 2% of the total data already achieves a comparable goal-oriented attack success rate (66.5%). Moreover, if the attacker’s only objective is to cause the VLA to fail, injecting just 1% of the data can increase the failure rate from 11.6% to 56.9% when the trigger is present.
>
> Suggestion 1:
>
> Based on our recent exploration, we found that the proposed GoBA mainly depends on the capability of the VLA. In this work, we focus on designing backdoor patterns that can successfully attack VLAs with limited ability.
>
> Suggestion 2:
>
> Thank you for the suggestion. We will consider it in future work. In this work, we assume a restricted scenario where the attacker cannot access the model and may not even know which model the developer will use.
>
> Suggestion 3:
>
> We will fix this.

---

> > ### Comment · Reviewer_Q5fa · 2025-11-26
> > **Official Comment by Reviewer Q5fa**
> >
> > I appreciate the detailed comments by the author. However, the response does not address my major concern regarding novelty:
> >
> > **W1**: Novelty can be evaluated by (1) New challenges the paper solves when applying existing methodology to new domains, and (2) Breakthrough findings the paper discovers when applying the same existing methodology to new domains.
> >
> > The author’s response fails to demonstrate either (1) or (2). I acknowledge the application-wise novelty: first backdoor attacks to VLAs. However, there is no algorithmic or conclusion novelty since the method works "out of the box" without requiring significant VLA domain-specific adaptations.
> >
> > Furthermore, the authors reference prior works like Carlini & Terzis to justify the novelty. However, that work was significant because it revealed a surprising empirical discovery: models could be compromised with extremely low poisoning rates (e.g., 0.01%), fundamentally challenging the safety of web-scale training (e.g., CLIP). In contrast, the results here are not surprising; showing that a 1-10% injection rate is effective aligns with established findings in other domains. Additionally, while the study on color/size effects is appreciated, it is domain-specific and does not offer broader insights to the research community.
> >
> > **W2**: Thanks for your reply. Please refine the threat model section accordingly.
> >
> > **Suggestion 1**: The reply doesn't address my concern. The Line 307 statement about VLA vulnerability from architectural aspects is still not rigorously supported, and I suggest removing it.
> >
> > **Suggestion 2**: The concern regarding technical/finding novelty remains unaddressed.

---

### Official Review · Reviewer_4JbT · 2025-10-30

**Soundness:** 3
**Presentation:** 2
**Contribution:** 2
**Rating:** 4
**Confidence:** 4

**Summary:**

This paper introduces the concept of goal-oriented backdoor attacks (GoBA) on vision-language-action (VLA) models, where adversaries inject physical objects as triggers into training data. Unlike traditional backdoor attacks that cause generic task failures, GoBA enforces specific actions only when the trigger is present, without degrading clean input performance. Leveraging the LIBERO benchmark, the authors present BadLIBERO and employ a three-level evaluation framework (nothing to do, try to do, success to do).

**Strengths:**

1. Addresses a practical security threat for embodied AI by showing attacks that can be implemented via physical objects, not just digital manipulations.

2. Clearly shows that VLAs, when poisoned via physical triggers, can reliably execute adversary-defined behaviors in real-world settings.

3. Proposes BadLIBERO, extending an established benchmark, facilitating reproducibility and further work. Introduces a detailed evaluation methodology, offering nuanced insights into attack efficacy.

4. Provides in-depth exploration of factors (trajectory, color, size) that influence attack effectiveness.

**Weaknesses:**

1. Related work is not sufficient enough. The paper does not provide enough background on vision-language-action (VLA) tasks, VLA models, or adversarial attacks in this domain. Specific models used are not properly introduced, leaving readers without essential context to judge the paper’s contribution.

2.  The work appears to largely re-use existing demonstrations from LIBERO with additional demonstrations, making the contribution seem closer to a dataset extension than an algorithmic or methodological advance. The novelty and distinct technical contributions should be clarified, ideally by reworking Section 1 or adding a dedicated subsection.

3. The manuscript’s organization is weak—section 3.1 may belong in section 2, and section 3.3 is poorly illustrated. Most importantly, there is a lack of detail regarding how GoBA is trained/finetuned and what occurs at inference. The methodology requires clearer exposition for readers to understand the proposed pipeline.

4. While the experimental breadth is appreciated, there is insufficient analysis of findings. For example, Section 5.4’s discussion of object difficulty (e.g., the “knife”) is not clearly linked to the main attack mechanism. I understand that the knife is hard to pick up so it fails more. But what message trying to convey here, how does that corresponding to the proposed GoBA?

5. The newly introduced 3-level evaluation metric is confusing, particularly at level-2. It is unclear how the attack success rate (ASR) is defined or what constitutes “success” at this stage (“trying to do what?”). A more intuitive explanation of the metric’s semantics is needed.

**Questions:**

1. The multiple-trigger experiments performed using identical triggers (e.g., the “same box of cookie”), what is the outcome if different triggers are used—does performance generalize, or is it highly dependent on the visual specifics of the trigger?

2. How would the proposed method extend to other benchmarks or tasks beyond LIBERO—does the attack depend on dataset idiosyncrasies or the embodied task definition?

3. What is the precise procedural pipeline for training and inference in GoBA? Does the attack require extra finetuning, or is it a plug-and-play poisoning method?

---

> ### Author Response · Authors · 2025-11-20
>
> Thank you for the valuable comments.
>
> W1:
>
> Regarding related work, we provide an overview of prior methods and their limitations in Section 1 (Lines 045–050). Specifically, existing attacks suffer from the following drawbacks:
>
> 1. Untargeted attacks: These methods cause general VLA failures rather than inducing specific malicious actions.
>
> 2. Digital triggers: Their triggers only exist in 2D images rather than in the 3D physical world, making them easily detectable and difficult to transfer to real-world scenarios.
>
> 3. Access rights: Many approaches require access to the victim model, which is often unrealistic.
>
> More detailed comparisons between prior work and our method are presented in Section 2 (Lines 110–134) and Table 1. A visual comparison is also provided in Figure 1, which clearly highlights the differences between our approach and existing methods.
> Regarding the model assumptions: in our attack setting, we do not assume any access to the victim model. Our focus is solely on modifying the training dataset, treating the model as a black box. This assumption is explicitly described in Section 3.2 (Lines 174–184). For the victim model used in our experiments, we cite the original paper appropriately (Lines 275–277).
>
> W2:
>
> Our work is a data-poisoning–based attack method that uses LIBERO (the most widely adopted benchmark in VLA domain) as the victim dataset. Our goal is to provide insights into how to create malicious samples in the VLA domain, and this work is not an extension of LIBERO.
>
> The key contributions of our work are as follows:
> 1. Goal-oriented (targeted) attack. To the best of our knowledge, this is the first method that enables an attacker to manipulate a VLA model such that the robot completes a specific attacker’s goal, rather than just causing task failure.
> 2. Physical trigger. Our trigger is an object, making the attack more realistic and significantly harder to detect compared to prior patch-based trigger approaches.
> 3. Minimal access requirement. Our method does not require access to the victim model. The attacker only needs to inject a small number of poisoned samples into the training dataset, making the attack easier to carry out in realistic settings.
>
> W3:
>
> As clarified in our response to W1, our attack does not rely on controlling or modifying the victim VLA’s training or fine-tuning procedure. Our core assumption is simply that the victim model is trained on the attacker-modified dataset. Therefore, we do not need to intervene in, or even know, the internal training pipeline of the victim models.
>
> For the experiments, we strictly follow the official training setup of the victim VLA, without any changes to its hyperparameters, architecture, or optimization process. The only difference is that the model is trained on our poisoned version of the dataset, rather than the original LIBERO dataset.
>
> W4:
>
> The core objective of the proposed GoBA model is to design harmful behaviors for the real world. Section 5.4 attempts to elucidate this phenomenon—in other words, under unreliable VLA control scenarios, having a robot pick up a knife constitutes a real-world hazardous action. While intuition is important, experimental results are still required for validation. Therefore, when attempting to implement goal-oriented backdoor attacks through data poisoning, the difficulty of achieving the target should be minimized as much as possible under conditions of limited data availability.
>
> W5:
>
> The measurement method involves dividing the number of instances falling into these three categories during test runs by the total number of test attempts. The level-2 “try to do” is the robot tries to follow the backdoor goal but fails. We split this into two cases It successfully picks up the triggered object but fails to place it on the attacker’s target surface.  And another case is trying to pick up the trigger but failing to grasp it.

---

> > ### Author Response · Authors · 2025-11-20
> >
> > Q1:
> >
> > Our multi-trigger experiments aim to demonstrate that when multiple same object (i.e., fake triggers) are placed near the real trigger, the VLA becomes confused and hesitates in deciding which object to pick up.
> >
> > For different triggers, as shown in Appendix B.1 and B.2, we observe that a backdoor trained with one specific trigger cannot be activated by another trigger that differs in packaging or size.
> >
> > However, we found one exception: a backdoored VLA trained with the original colorful packaging can still be activated by the same object wrapped in Gaussian-noise packaging. This suggests that, for this particular case, the VLA fails to distinguish between the original packaging and the Gaussian-noise–altered version, implying that the model considers them visually similar enough to match the learned trigger pattern.
> >
> > Q2:
> >
> > We believe this kind of method can extend to any datasets in the VLA domain, only if the attacker can participate in the data collection process. We try this attack on the SOTA method OpenVLA-OFT, for which we can see the highest attack success rate, we believe this method depends on the VLA’s learning ability—the stronger the learning ability, the better the attacker's performance.
> >
> > Q3:
> >
> > As demonstrated in our response to W3, our attack does not require designing specific training or fine-tuning processes for VLA, making it a plug-and-play poisoning method.

---

> > > ### Comment · Reviewer_4JbT · 2025-11-26
> > >
> > > Thank you for the authors’ response. However, I remain unconvinced by the clarification regarding the claimed novelty. As also pointed out by other reviewers, the proposed approach appears to be an adaptation of well-established backdoor attack methods, merely applied to the VLA setting. The experiments are not sufficiently designed to highlight any unique challenges or characteristics specific to VLA backdoor attacks, and the paper fails to provide new insights into this area. Therefore, I will keep my original score.

---

### Meta-Review · Area_Chair_vfVx · 2026-01-07

**Summary:**

This paper presents goal-oriented backdoor attacks (GoBA), a backdoor attack method against VLA models via physical triggers. The reviewers mostly acknowledged the importance of studying security threats to VLA models in physical environments and the relevance of considering physical triggers. They also raised major concerns on the technical novelty and depth of the work, commenting that the attack is a relatively straightforward adaptation of traditional backdoor attacks in VLA context and there is a lack of in-depth study on VLA-specific challenges.

**Reviewer Concerns:**

The authors provided rebuttal to all the comments. Reviewers 4JbT, Q5fa, and GP82 responded that they still have concerns on the work's novelty and depth. Reviewer EhrE also has concerns on limited novelty and did not respond.

**Reviewer Scores:**

Three reviewers responded and will likely maintain their scores (4, 2, 2). One reviewer did not respond and is unlikely to change their score (4).

---

### Decision · Program_Chairs · 2026-01-26

Reject